# Serum NGF and BDNF in Long-COVID-19 Adolescents: A Pilot Study

**DOI:** 10.3390/diagnostics12051162

**Published:** 2022-05-07

**Authors:** Carla Petrella, Raffaella Nenna, Laura Petrarca, Francesca Tarani, Roberto Paparella, Enrica Mancino, Greta Di Mattia, Maria Giulia Conti, Luigi Matera, Enea Bonci, Flavio Maria Ceci, Giampiero Ferraguti, Francesca Gabanella, Christian Barbato, Maria Grazia Di Certo, Luca Cavalcanti, Antonio Minni, Fabio Midulla, Luigi Tarani, Marco Fiore

**Affiliations:** 1Institute of Biochemistry and Cell Biology (IBBC-CNR), Department of Sensory Organs, Sapienza University of Rome, 00185 Roma, Italy; carla.petrella@cnr.it (C.P.); francesca.gabanella@cnr.it (F.G.); christian.barbato@cnr.it (C.B.); mariagrazia.dicerto@cnr.it (M.G.D.C.); 2Department of Maternal Infantile and Urological Sciences, Sapienza University of Rome, 00185 Roma, Italy; raffaella.nenna@uniroma1.it (R.N.); laura.petrarca@uniroma1.it (L.P.); francesca.tarani@uniroma1.it (F.T.); roberto.paparella@uniroma1.it (R.P.); enrica.mancino@uniroma1.it (E.M.); greta.dimattia@uniroma1.it (G.D.M.); mariagiulia.conti@uniroma1.it (M.G.C.); luigi.matera@uniroma1.it (L.M.); fabio.midulla@uniroma1.it (F.M.); luigi.tarani@uniroma1.it (L.T.); 3Department of Experimental Medicine, Sapienza University of Rome, 00185 Roma, Italy; enea.bonci@uniroma1.it (E.B.); flaviomaria.ceci@uniroma1.it (F.M.C.); giampiero.ferraguti@uniroma1.it (G.F.); 4Department of Sensory Organs, Sapienza University of Rome, 00185 Roma, Italy; luca.cavalcanti@uniroma1.it (L.C.); antonio.minni@uniroma1.it (A.M.)

**Keywords:** neurotrophins, NGF, BDNF, adolescents, long-COVID-19, biomarkers, SARS-CoV-2

## Abstract

COVID-19 (COronaVIrus Disease 19) is an infectious disease also known as an acute respiratory syndrome caused by the SARS-CoV-2. Although in children and adolescents SARS-CoV-2 infection produces mostly mild or moderate symptoms, in a certain percentage of recovered young people a condition of malaise, defined as long-COVID-19, remains. To date, the risk factors for the development of long-COVID-19 are not completely elucidated. Neurotrophins such as NGF (Nerve Growth Factor) and BDNF (Brain-Derived Neurotrophic Factor) are known to regulate not only neuronal growth, survival and plasticity, but also to influence cardiovascular, immune, and endocrine systems in physiological and/or pathological conditions; to date only a few papers have discussed their potential role in COVID-19. In the present pilot study, we aimed to identify NGF and BDNF changes in the serum of a small cohort of male and female adolescents that contracted the infection during the second wave of the pandemic (between September and October 2020), notably in the absence of available vaccines. Blood withdrawal was carried out when the recruited adolescents tested negative for the SARS-CoV-2 (“post-infected COVID-19”), 30 to 35 days after the last molecular test. According to their COVID-19 related outcomes, the recruited individuals were divided into three groups: asymptomatics, acute symptomatics and symptomatics that over time developed long-COVID-19 symptoms (“future long-COVID-19”). As a control group, we analyzed the serum of age-matched healthy controls that did not contract the infection. Inflammatory biomarkers (TNF-α, TGF-β), MCP-1, IL-1α, IL-2, IL-6, IL-10, IL-12) were also analyzed with the free oxygen radicals’ presence as an oxidative stress index. We showed that NGF serum content was lower in post-infected-COVID-19 individuals when compared to healthy controls; BDNF levels were found to be higher compared to healthy individuals only in post-infected-COVID-19 symptomatic and future long-COVID-19 girls, leaving the BDNF levels unchanged in asymptomatic individuals if compared to controls. Oxidative stress and inflammatory biomarkers were unchanged in male and female adolescents, except for TGF-β that, similarly to BDNF, was higher in post-infected-COVID-19 symptomatic and future long-COVID-19 girls. We predicted that NGF and/or BDNF could be used as early biomarkers of COVID-19 morbidity in adolescents.

## 1. Introduction

COVID-19 (COronaVIrus Disease 19), is an infectious disease also known as an acute respiratory syndrome caused by the SARS-CoV-2 (Severe Acute Respiratory Syndrome Coronavirus 2) [1,2,3,4,5]. The introduction of vaccination in early 2021 decreased the pressure on health systems [6,7,8], significantly reducing the number of hospitalized people. However, the question of the diffusibility of the virus remains open, especially among children not subjected to vaccination. It is evident that in children and adolescents the SARS-CoV-2 infection, when not asymptomatic, produces mostly symptoms of mild or moderate intensity [9,10,11,12]. It is also well established that in a certain percentage of recovered young people a condition of malaise, defined as long-COVID-19, remains, a long-term consequence of SARS-CoV-2 infection [13,14,15,16,17,18,19].

Neurotrophins (NT) are growth factors known to regulate neuronal growth, survival and morphology during development and in the adult brain [20,21,22]; they are also able to influence important functions such as excitability, synaptogenesis and brain aging [23,24]. The NT family includes the NGF (Nerve Growth Factor), BDNF (Brain-Derived Neurotrophic Factor), NT-3, NT-4, NT-5, NT-6, and NT-7 with neurotrophic and neuroprotective activity on different neuronal populations, both in the peripheral nervous system and in the central nervous system. However, NGF and BDNF are active not only in nerve cells, but also play a key role as endocrine and paracrine regulators of the cardiovascular, immune, and endocrine systems to regulate homeostasis in physiological and/or pathological conditions [25,26,27,28,29,30]. As for the relationship between COVID-19 and NT, to date only a few papers have discussed their potential role in this disease. Previous studies debated the role of NGF in pulmonary pathologies, alluding to the possibility of considering NGF signaling as a potential diagnostic/therapeutic target in SARS-CoV-2 induced-pulmonary complications [31] contributing to antibody production in convalescent COVID-19 individuals [32]. Other investigations proposed that serum BDNF content and BDNF/adiponectin ratio may serve as predictors of worsened prognosis in COVID-19, especially for adult male patients [33], with BDNF also playing a subtle role in the neurological and mental outcomes of COVID-19 patients [34,35].

To the best of our knowledge, no findings are available on the NGF and BDNF serum levels in adolescents affected by SARS-CoV-2. Thus, the main aims and novelties of the present preliminary COVID-19 study were to disclose changes in the NGF and BDNF concentrations in the serum of adolescents. We also measured inflammatory biomarkers such as Tumor Necrosis Factor-α (TNF-α), Transforming Growth Factor-β (TGF-β), Monocyte Chemoattractant Protein-1 (MCP- 1), Interleukin (IL)-1α, IL-2, IL-6, IL-10, IL-12; and oxidative stress, analyzed as free oxygen radicals defense (FORD) and using the free oxygen radicals test (FORT), in the serum of the recruited adolescents. We predicted that neurotrophins NGF and BDNF could be used as early biomarkers of long-COVID-19 morbidity in adolescents. 

## 2. Results

The analysis of age, body weight and BMI of the enrolled SARS-CoV-2 individuals and healthy negative adolescents did not disclose differences between COVID-19 groups, but only the expected gender effect on body weight (data not shown).

Figure 1A,B displays the NGF and BDNF serum levels in male and female post-infected-COVID-19 adolescents and the control groups. ANOVA data for NGF (Figure 1A) clearly disclose the main COVID-19 and gender effects (F(3,32) = 5.99, *p* < 0.001; F(1,32) = 12.44, *p* < 0.01, respectively) and an interaction COVID-19 × gender (F(3,32) = 5.73, *p* < 0.01). In particular, post hoc comparisons evidenced reduced levels of NGF in acute symptomatic post-infected-COVID-19 and future long-COVID-19 boys and girls compared to healthy individuals (ps < 0.05 or less). However, this NGF decrease was also observed for post-infected-COVID-19 asymptomatic girls. 

Figure 1B shows that COVID-19 exposure elicited elevated levels of BDNF in post-infected-COVID-19 acute symptomatic and future long-COVID-19 girls. Indeed, ANOVA data revealed an interaction, COVID-19 x gender, in the BDNF analysis (F(3,32) = 3.48, *p* < 0.05; post hoc, ps < 0.05).

Figure 2A,B shows the oxidative stress status of the recruited individuals measured as FORT and FORD (see Methods). ANOVA investigation did not indicate COVID-19 or gender effects, nor interactions between COVID-19 and gender (data not shown).

Figure 3 and Figure 4 display the ANOVA data on IL-1α, IL-2, IL-6, IL-10, IL-12, MCP-1, TGF-β, and TNF-α of male and female post-infected COVID-19 adolescents compared to healthy boys and girls used as controls. Statistical analysis did not show COVID-19 or gender effects or interactions for IL-1α, IL-2, IL-6, IL-10, IL-12, MCP-1, and TNF-α. However, COVID-19 potentiated TGF-β in acute symptomatic and future long-COVID-19 girls compared to healthy girls. Indeed, although in the absence of significant ANOVA data for COVID-19, gender or interaction COVID-19 x gender, the Tukey’s post hoc comparisons (the use of which is acceptable or even suggested also without significant ANOVA main or interaction effects [36]) disclosed high levels of TGF-β according to COVID-19 morbidity in girls (ps < 0.05 in post hoc comparisons).

The Spearman correlation tests did not evidence effects on BMI and age in the considered parameters for all groups.

**Figure 1 diagnostics-12-01162-f001:**
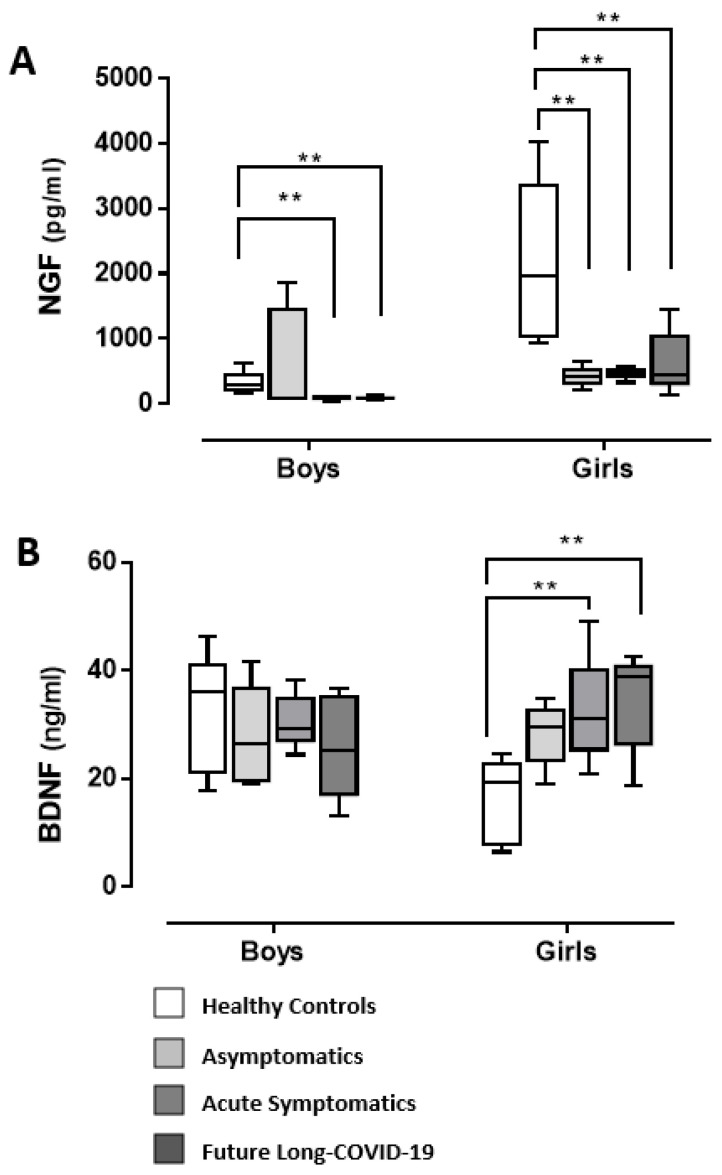
Serum NGF (**A**) and BDNF (**B**) in COVID-19 adolescents and healthy controls. According to their COVID-19 related outcomes, the recruited individuals were divided into 3 groups: asymptomatics, acute symptomatics, and symptomatics that over time developed long-COVID-19 symptoms (“future long-COVID-19”); as a control group we analyzed the serum of age-matched healthy controls that did not contract the infection. Boxes indicate the lower and upper quartiles. The vertical lines extending from each box represent the minimum and maximum values. The asterisks indicate significant differences between groups (** *p* < 0.01).

**Figure 2 diagnostics-12-01162-f002:**
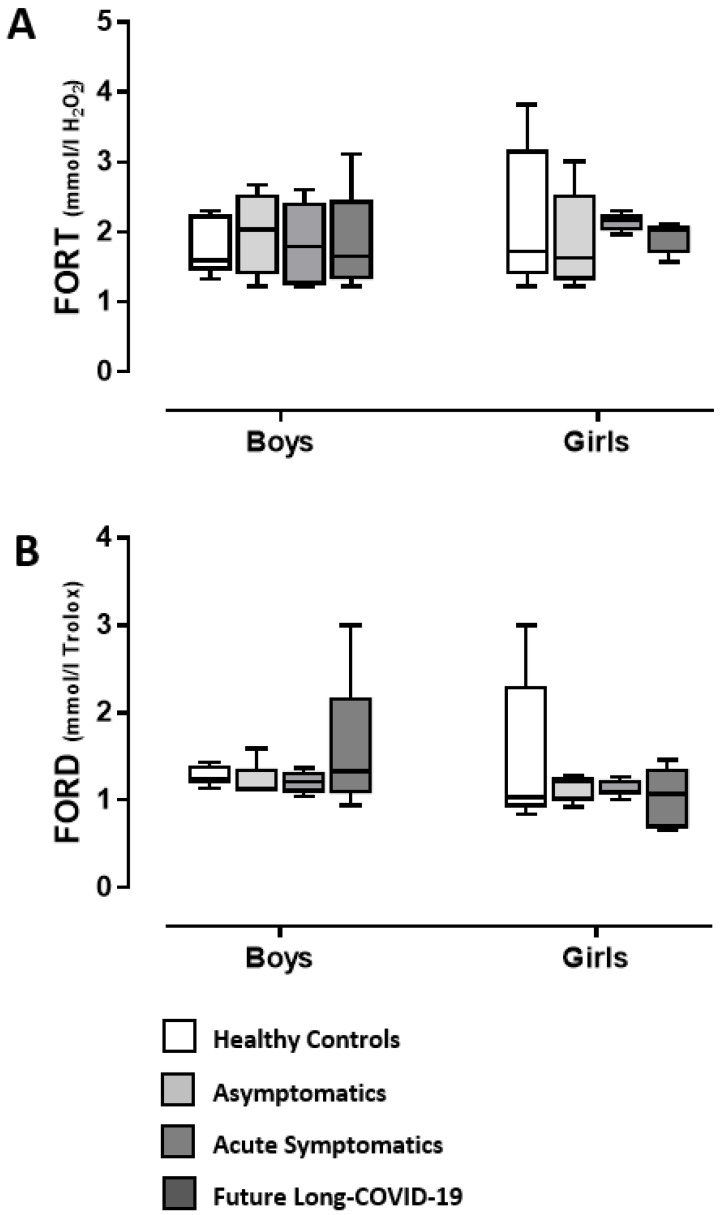
Serum FORT (**A**) and FORD (**B**) in COVID-19 adolescents and healthy controls. According to their COVID-19 related outcomes, the recruited individuals were divided into 3 groups: asymptomatics, acute symptomatics, and symptomatics that over time developed long-COVID-19 symptoms (“future long-COVID-19”); as a control group we analyzed the serum of age-matched healthy controls that did not contract the infection. Boxes indicate the lower and upper quartiles. The vertical lines extending from each box represent the minimum and maximum values.

**Figure 3 diagnostics-12-01162-f003:**
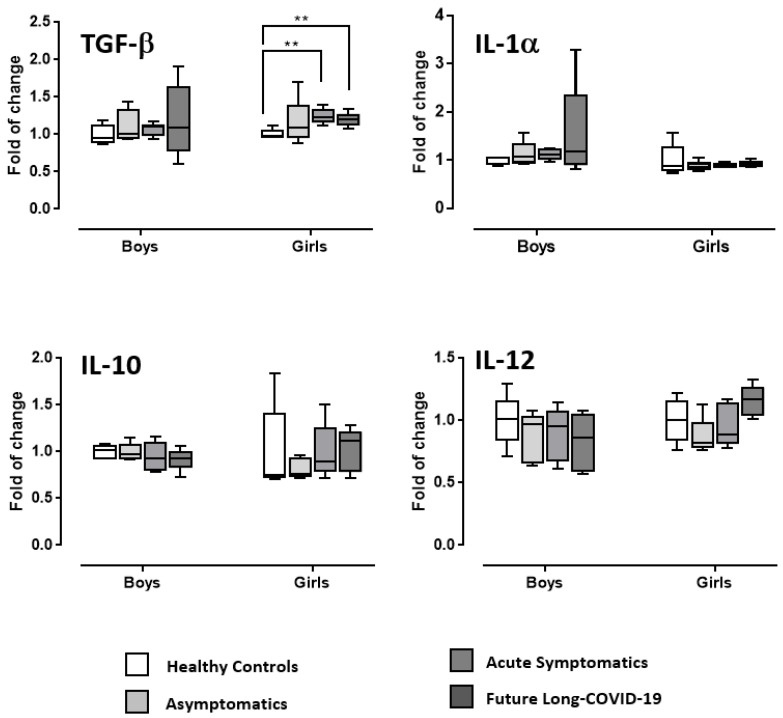
Serum TGF-β, IL-1α, IL-10, and IL-12 in COVID-19 adolescents and healthy controls. According to their COVID-19 related outcomes, the recruited individuals were divided into three groups: asymptomatics, acute symptomatics, and symptomatics that over time developed long-COVID-19 symptoms (“future long-COVID-19”); as a control group we analyzed the serum of age-matched healthy controls that did not contract the infection. Boxes indicate the lower and upper quartiles. The vertical lines extending from each box represent the minimum and maximum values. The asterisks indicate significant differences between groups (** *p* < 0.01).

**Figure 4 diagnostics-12-01162-f004:**
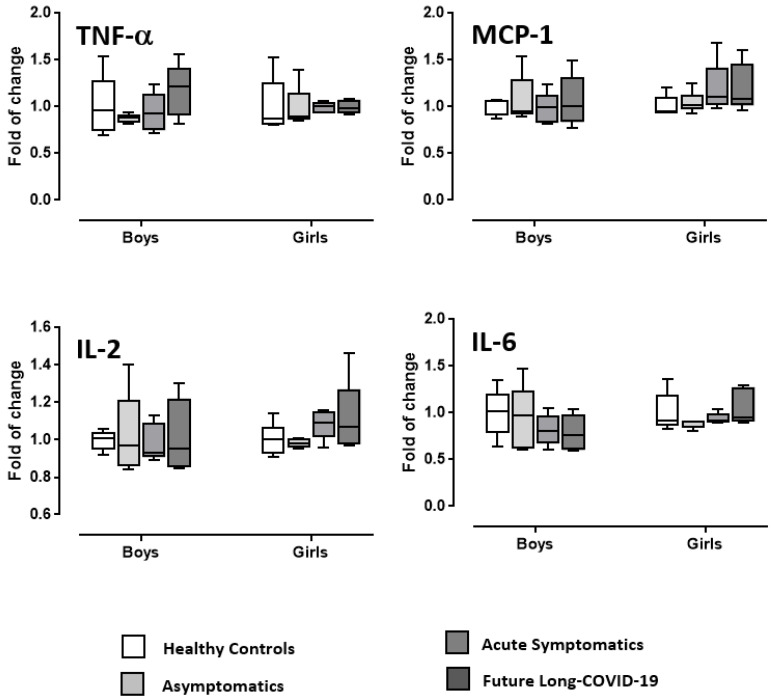
Serum TNF-α, MCP-1, IL-2, and IL-6 in COVID-19 adolescents and healthy controls. According to their COVID-19 related outcomes, the recruited individuals were divided into 3 groups: asymptomatics, acute symptomatics, symptomatics that over time developed long-COVID-19 symptoms (“future long-COVID-19”); as a control group we analyzed the serum of age-matched healthy controls that did not contract the infection. Boxes indicate the lower and upper quartiles. The vertical lines extending from each box represent the minimum and maximum values.

## 3. Discussion

This is the first study evaluating the role of NGF and BDNF in a small cohort of male and female adolescents one month after testing negative for COVID-19 (post-infected COVID-19), compared to healthy controls. In particular, we found differences between the post-infected-COVID-19 subgroups considered (asymptomatic, acute symptomatic and symptomatic that had maintained long-lasting symptoms after negativization/future long-COVID-19) compared to healthy adolescents, and a gender-specific serum neurotrophin profile. In fact, whilst NGF serum content in both girls and boys was lower in post-infected COVID-19 individuals when compared to healthy controls, BDNF levels were found to be higher compared to normal conditions only in post-infected-COVID-19 symptomatic girls and future long-COVID-19 girls; BDNF levels were unchanged in asymptomatic individuals when compared to controls. The same individuals did not show any modified serum oxidative stress, considering both the reactive oxygen species levels (FORT test) and the endogenous anti-oxidant system functionality (FORD test). Concerning the inflammatory condition, one month after negativization there was a substantial absence of altered levels of some of the cytokine/chemokines considered (TNF-α, IL-1α, MCP-1, IL-2, IL-6, IL-10, IL-12). Quite interestingly, TGF-β serum levels were higher than in healthy controls only in symptomatic post-infected-COVID-19 and future long-COVID-19 girls. Long-COVID-19 symptomatology involved both adults and young patients, with symptoms for more than 6 months from the end of the acute phase of the disease or the negativization of the swab [37,38,39,40]. The main symptoms of COVID-19 are fatigue, asthenia, cognitive dysfunction, and shortness of breath, but also include effects on the central nervous, muscle-skeletal, and cardiovascular systems, impacting everyday functioning. The symptoms may disappear and then reappear, even after months, perhaps as a result of even minimal physical or psychological stress [41]. To date, risk factors for the development of long-COVID-19 are not completely elucidated. This could be due to its multiple symptomatic appearances, oscillating from long-term multi-organ damage to unresolved inflammation from multiple sources [17,42,43,44,45]. Currently, there are data, divided by sex, on the incidence of long-COVID-19 only in the adult population. Quite interestingly, long-COVID-19 appears to be more probable in women than in men [46] in contrast to severe acute COVID-19 [47,48], independent of the severity of the original infection. Other risk factors for having long-COVID-19 seem to be a higher body mass index and older age [49,50,51].

We found that NGF serum levels in post-infected-COVID-19 adolescents were decreased compared with healthy controls both in boys and girls (except for asymptomatic male individuals who showed unchanged NGF levels). A few scientific papers underpinned the relationship between NGF and SARS-CoV-2 infection. A recent study revealed that biological fluid levels of NGF (the milk of mothers with a confirmed COVID-19 PCR test) were lower than in healthy mothers, suggesting that SARS-CoV-2 infection could influence the secretion of NGF. The impact of this change on newborns’ neurodevelopment could be the subject of intriguing future studies [52]. A recent study showed that the β-NGF/TrkA signaling pathway is associated with the production of anti-nucleoprotein IgG in convalescent COVID-19 individuals with specific effects on the virus-humoral and T cell responses [32]. Interestingly, both NGF and stress conditions are intimately connected. The persistent alteration of the stress axis pathologically changes physiological balances, and this is the basis of dysfunctions at different levels [53,54]. Evidence clearly shows NGF’s role in the modulation of the hypothalamic-pituitary-adrenal (HPA) axis in response to stressful stimuli, correlating environmental stimuli with pathological and physiological feedbacks [55,56], as demonstrated in human and animal model studies [25,57,58]. In conditions of chronic stress and pathological conditions such as depression, the chronic hyperactivity of the HPA axis is associated with cortical atrophy, leading to long-term NGF downregulation [59].

In the context of COVID-19, an important consideration, in no way secondary to clinical symptoms, is the impact of the social isolation to which the population has been subjected, with lasting psycho-social consequences over time [60]. Quarantine and social distancing—protection measures from the COVID-19 pandemic—have represented potential sources of stress for children and adolescents, precisely because of the persistence of sudden and prolonged changes in the daily rhythms of family and school life (loss of routine and reduction in educational and playful/exploratory outdoors activities) and of “breathing” a climate of anxiety/fear and uncertainty for the future [61,62]. In light of these considerations, we could speculate that an NGF decreased serum level reflects a persistent activation of the stress axis, underlying an unknown mechanism. This effect could be a direct effect of the virus, which can reach the central nervous system through the olfactory system [63], or an indirect consequence of psychological impairment due to the physical and social consequences of the pandemic. 

We found a remarkable difference in BDNF levels; only post-COVID-19 symptomatic and future long-COVID-19 girls showed increased BDNF serum content, compared to healthy controls. The elevated BDNF levels correspond to an increase in TGF-β in the serum of the same patients. In fact, one month after the disappearance of the virus, despite an inflammatory and oxidative stress picture in line with the healthy controls, the serum level of this specific cytokine remains augmented only in symptomatic and future long-COVID-19 girls. 

TGF-β regulates numerous biological processes including lung organogenesis and homeostasis [64,65]. BDNF has also been identified as a mediator of bronchial hyper-responsiveness, and its augmented release by platelets and mononuclear cells in the presence of lower respiratory tract infections has been demonstrated in a pilot study [66]. Interestingly, both BDNF and TGF-β mediators are stored in platelets and released in inflammatory conditions, as shown in the serum of patients affected by Chronic Obstructive Pulmonary Disease [67]. As for COVID-19, TGF-β has been proposed as an attractive target for therapeutic intervention because of its pro-fibrinogenic and immunosuppressive effects that are elevated during and after COVID-19 [68,69]. We speculate that the persistent elevation in BDNF and TGF-β serum levels in post-COVID-19 girls that experienced respiratory symptoms during the acute phase of the infection could represent an alarm bell for the long-term effects of COVID-19. 

A main point of the present COVID-19 pilot investigation is a predicted lack of differences in most of the inflammatory cytokines analyzed from adolescents. We do speculate that this absence of effects could be due to two main reasons: (i) young people, luckily, are less exposed to the deleterious effects of the SARS-CoV-2; (ii) the blood withdrawal was carried out at least 30 days after negativization. 

As for gender differences, we found in NGF/BDNF serum presence that this crucial issue is a well-known aspect of the neurotrophins’ saga. Indeed, many factors may contribute to NGF and/or BDNF gender differences in humans: namely, stressful situations, the hour of the day/night, hormonal variations, age, food intake, and personal psychological profiles [70,71,72,73].

The strength and novelty of this study were to score early biomarkers in COVID-19 adolescents according to their final prognosis based (or not) on long-term COVID-19 effects during the second pandemic paroxysm wave and in absence of vaccines. However, the relatively small number of adolescents recruited for this study may represent a possible limit of the study; on the other hand, its strength depends on the adoption of quite restricted enrollment rules.

## 4. Materials and Methods

### 4.1. Adolescents’ Recruitment 

COVID-19 adolescents (positive at a PCR test) recruited to our study included 30 boys and girls (see Table 1 and Table 2, age-range 13–16 years) in follow-up at the Department of Maternal Infantile and Urological Sciences of the Sapienza University Hospital “Policlinico Umberto I” of Rome, Italy. Blood withdrawal was carried out when the adolescents were negative for SARS-CoV-2 (“post-infected COVID-19”), 30 to 35 days after the PCR test. According to their COVID-19 related outcomes, recruited individuals were divided into 3 groups: asymptomatics, acute symptomatics, symptomatics that over time developed long-COVID-19 symptoms (“future long-COVID-19”) (*n*: 5 boys and 5 girls for each group); as a control group (5 boys and 5 girls, age range 13–17 years). We analyzed the serum of age-matched healthy controls that did not contract the infection by both anti-spike and anti-nucleoprotein blood analyses. We recruited these adolescents because they attended the hospital for the investigation of presumed pathologies that were not present at all, thus defining the adolescents as “healthy” [74,75,76].

For the adolescents’ recruitment, several pieces of information (when available) were obtained [76]: physical examination and anthropometric parameters measurements including gender, ethnicity, weight, height, and physiological anamnesis; family history including diseases, parents‘ age at pregnancy, parents’ education; pharmacological anamnesis; and close and remote pathological anamnesis. As previously shown [76], the main exclusion criteria used to avoid bias in the selection of adolescents (including controls) at the beginning of recruitment to a much larger cohort included: other ongoing pathologies; previous inflammatory, endocrine and autoimmune disorders; diagnosed cardiovascular pathologies that could have biased inflammatory analysis; previous use of drugs or chemicals that can alter the serum levels of inflammation markers, such as antidepressants, anti-inflammatories and immunosuppressants.

The study was approved by the Sapienza University Hospital ethical committee (Ref. 0399/2021); an informed consent was signed by each parent of every adolescent and all study procedures were compliant with the Helsinki Declaration of 1975, as revised in 1983, for human experimentation.

### 4.2. Blood Withdrawal

According to methods previously described [76], peripheral blood samples of 5 mL were taken from each participant, collected in BD Vacutainer™ Serum Separation Tubes and centrifuged at 3000 rpm for 15 min to separate serum. Serum was then stored at −80 °C.

### 4.3. NGF and BDNF Serum Level Evaluation

NGF (Cat. No. DY256) and BDNF (Cat. No. DY248) were measured using sandwich enzyme-linked immunosorbent assay (ELISA) kits (R&D Systems, Minneapolis, MN, USA), according to the protocols provided by the manufacturer and also according to methods previously described [25,77]. Serum samples were diluted 2- and 100-fold with PBS for detection of NGF and BDNF, respectively. The colorimetric reaction product was measured at 450 nm using a microplate reader (Neo Biotech Microplate Reader, Milan, Italy). Data are represented as ng/mL (BDNF) or pg/mL (NGF) and all assays were performed in duplicate which was averaged for statistical comparison [76].

### 4.4. Free Oxygen Radicals Defense (FORD) and Free Oxygen (FORT) Serum Evaluation

FORD and FORT tests were carried out using two specific kits (both purchased from Callegari, Parma, Italy) following the instructions provided by the manufacturer and according to methods previously described [25,76]. The FORD test facilitates the determination of free oxygen radicals’ defense. Briefly, this test uses a preformed stable and colored radical and determines the decrease in absorbance that is proportional to the antioxidant concentration of the sample [78]. The FORT test allows the determination of free oxygen radicals (ROS) through a colorimetric assay based on the ability of transition metals, such as iron, to catalyze the breakdown of hydroperoxides (ROOH) into derivative radicals, according to Fenton’s reaction [78]. 

### 4.5. Oxidative Stress ELISA Strip Profiling Assay 

In this study, we used Human Oxidative Stress ELISA Strip for Profiling 8 Cytokines (Catalog Number EA-1301, Signosis, Santa Clara, CA, USA) that simultaneously analyzes the following cytokines: TNF-α, TGF-β, MCP-1, IL-1α, IL-2, IL-6, IL-10, and IL-12. 

Each well of the strip is coated with a specific capture antibody to detect its corresponding cytokine in the sample. Therefore, 8 different proteins can be measured simultaneously. The test sample reacts simultaneously with pairs of two antibodies, resulting in the cytokines being sandwiched between the solid phase and enzyme-linked antibodies. After incubation, the wells are washed to remove unbound-labeled antibodies. The HRP substrate, TMB, is then added which causes the color to change to blue. The reaction is then terminated with Stop Solution, resulting in a yellow color. The concentrations of cytokines are directly proportional to the color intensity of the test sample. Absorbance is measured spectrophotometrically at 450 nm, within 30 min. The results were expressed as a fold of change compared to control values.

### 4.6. Statistical Analysis 

According to methods previously described [79,80], data were analyzed to assess normality by Pearson’s chi-squared test and two-way analysis of variance (ANOVA) (controls vs. asymptomatics vs. acute symptomatics vs. acute symptomatics that over time developed long-COVID-19 symptoms and boys vs. girls) was used to analyze the laboratory parameters. The Bonferroni correction was used to counteract the multiple comparisons problem for the NGF/BDNF analyses, the cytokines’ investigations and the FORD/FORT studies. Post hoc comparisons were carried out using Tukey’s HSD test. The Spearman correlation test was used to investigate the correlation between the laboratory data and the age and BMI of the patients. 

## 5. Conclusions

In the effort to unravel other primary potential COVID-19 biomarkers [1,81], this pilot study provides additional information aimed at disclosing further biomolecular events consequent to SARS-CoV-2 infection. Particularly, serum BDNF could represent a new tool as an early predictor of COVID-19 long-term effects, especially in girls.

## Figures and Tables

**Table 1 diagnostics-12-01162-t001:** Symptoms during the acute phase of COVID-19 disease in all 40 adolescents enrolled in the study. The number of cases vs. total patients was reported for each symptom. According to their COVID-19 related outcomes, recruited individuals were divided into 3 groups: asymptomatics, acute symptomatics, and acute symptomatics that over time developed long-COVID-19 symptoms (“future long-COVID-19”); as a control group, we analyzed the serum of age-matched healthy controls that did not contract the infection.

GROUPS		Healthy Controls		Asymptomatics COVID-19		Acute Symptomatics COVID-19		Future Long-COVID-19	
		Boys (*n* = 5)	Girls (*n* = 5)	Boys (*n* = 5)	Girls (*n* = 5)	Boys (*n* = 5)	Girls (*n* = 5)	Boys (*n* = 5)	Girls (*n* = 5)
**SYMPTOMATOLOGY**									
Symptoms during acute COVID-19	Fever	-	-	0/5	0/5	2/5	4/5	1/5	3/5
	Cough	-	-	0/5	0/5	2/5	1/5	1/5	3/5
	Breathing difficulties	-	-	0/5	0/5	0/5	0/5	1/5	3/5
	Rhinitis	-	-	0/5	0/5	2/5	1/5	0/5	1/5
	Ear infection	-	-	0/5	0/5	0/5	0/5	0/5	1/5
	Pharyngitis	-	-	0/5	0/5	1/5	2/5	3/5	1/5
	Respiratory system	-	-	0/5	0/5	4/5	2/5	3/5	4/5
	Diarrhea	-	-	0/5	0/5	0/5	0/5	1/5	1/5
	Vomiting	-	-	0/5	0/5	0/5	1/5	2/5	0/5
	Nausea	-	-	0/5	0/5	0/5	0/5	0/5	0/5
	Abdominal pain	-	-	0/5	0/5	0/5	0/5	0/5	0/0
	Gastrointestinal pain	-	-	0/5	0/5	0/5	1/5	2/5	1/5
	Acute chest pain	-	-	0/5	0/5	0/0	0/5	1/5	1/5
	Asthenia	-	-	0/5	0/5	2/5	2/5	4/5	4/5
	Ageusia	-	-	0/5	0/5	4/5	2/5	2/5	3/5
	Anosmia	-	-	0/5	0/5	4/5	1/5	3/5	3/5
	Headache	-	-	0/5	0/5	3/5	5/5	2/5	3/5
	Neurological abnormalities	-	-	0/5	0/5	5/5	5/5	4/5	5/5
	Skin Rash	-	-	0/5	0/5	0/5	0/5	1/5	2/5
	Acute Musculoskeletal Symptoms	-	-	0/5	0/5	0/5	2/5	1/5	1/5
	Myalgia	-	-	0/5	0/5	0/5	1/5	1/5	1/5

**Table 2 diagnostics-12-01162-t002:** Symptoms after the acute phase of COVID-19 disease in all 40 adolescents enrolled in the study. The number of cases vs. total patients was reported for each symptom. According to their COVID-19 related outcomes, recruited individuals were divided into 3 groups: asymptomatics, acute symptomatics, and acute symptomatics that over time developed long-COVID-19 symptoms (future long-COVID-19); as a control group, we analyzed the serum of age-matched healthy controls that did not contract the infection.

GROUPS		Healthy Controls		Asymptomatics COVID-19		Acute Symptomatics COVID-19		Future Long-COVID-19	
		Boys (*n* = 5)	Girls (*n* = 5)	Boys (*n* = 5)	Girls (*n* = 5)	Boys (*n* = 5)	Girls (*n* = 5)	Boys (*n* = 5)	Girls (*n* = 5)
**SYMPTOMATOLOGY**									
Symptoms Post-COVID-19	Fever	-	-	0/5	0/5	0/5	0/5	-	-
	Cough	-	-	0/5	0/5	0/5	0/5	0/5	1/5
	Breathing difficulties	-	-	0/5	0/5	0/5	0/5	0/5	0/5
	Rhinitis	-	-	0/5	0/5	0/5	0/5	-	-
	Ear infection	-	-	0/5	0/5	0/5	0/5	-	-
	Pharyngitis	-	-	0/5	0/5	0/5	0/5	-	-
	Respiratory system	-	-	0/5	0/5	0/5	0/5	0/5	1/5
	Diarrhea	-	-	0/5	0/5	0/5	0/5	-	-
	Vomiting	-	-	0/5	0/5	0/5	0/5	-	-
	Nausea	-	-	0/5	0/5	0/5	0/5	-	-
	Abdominal pain	-	-	0/5	0/5	0/5	0/5	-	-
	Gastrointestinal pain	-	-	0/5	0/5	0/5	0/5	0/5	0/5
	Acute chest pain	-	-	0/5	0/5	0/5	0/5		-
	Asthenia	-	-	0/5	0/5	0/5	0/5	2/5	3/5
	Ageusia	-	-	0/5	0/5	0/5	0/5	1/5	1/5
	Anosmia	-	-	0/5	0/5	0/5	0/5	2/5	1/5
	Headache	-	-	0/5	0/5	0/5	0/5	2/5	1/5
	Neurological abnormalities	-	-	0/5	0/5	0/5	0/5	4/5	2/5
	Skin Rash	-	-	0/5	0/5	0/5	0/5	-	-
	Acute Musculoskeletal Symptoms	-	-	0/5	0/5	0/5	0/5	0/5	0/5
	Myalgia	-	-	0/5	0/5	0/5	0/5	0/5	0/5

## Data Availability

Data are available on request.

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
