# Peer review of "Serum NGF and BDNF in Long-COVID-19 Adolescents: A Pilot Study"

_diagnostics, 2022, doi:10.3390/diagnostics12051162_

Round 1
Reviewer 1 Report
It is much better to consider the study as a pilot.
The changes made in the manuscript answer my concerns.
Author Response
Answers to the comments of Reviewer 1
It is much better to consider the study as a pilot.
The changes made in the manuscript answer my concerns.
Reply: we do thank the reviewer for the positive comments
Reviewer 2 Report
the paper has been highly enhanced and I think my queries are answered,
the only concern is that the paper requires some English corrections:
please change all the Dont, didnt and It's into a formal English writing: do not, did not, It is etc...
as to references, there are seminal work correlating Covid, neuro symptoms and immune system, please cite in your work.
Author Response
Answers to the comments of Reviewer 2
the paper has been highly enhanced and I think my queries are answered,
reply: we do thank the reviewer for the positive comments
the only concern is that the paper requires some English corrections:
please change all the Dont, didnt and It's into a formal English writing: do not, did not, It is etc...
reply: as suggested we made changes (highlighted in light green)
as to references, there are seminal work correlating Covid, neuro symptoms and immune system, please cite in your work
reply: as requested, we included additional key references in the revised paper (highlighted in light green)
Reviewer 3 Report
Since the barplot and boxplot contain the same information, please remove the barplot from figures 1-4 and keep only box plots.
Author Response
Answers to the comments of Reviewer 3
Since the barplot and boxplot contain the same information, please remove the barplot from figures 1-4 and keep only box plots.
Reply: as requested, the figures are now only as box plots.
This manuscript is a resubmission of an earlier submission. The following is a list of the peer review reports and author responses from that submission.
Round 1
Reviewer 1 Report
Looking for early markers for covid-19 un adolescents, the authors measure the neurotrophins NGF (Nerve Growth Factor) and BDNF (Brain-Derived Neurotrophic Factor) that influence neuronal development maintenance and plasticity, as well as cardiovascular, immune, and endocrine.
They studied a small cohort of serum from male and female adolescents without vaccines that contracted the infection on the second pandemic wave of covid-19. They evaluated other immune system mediators like TGFb, some interleukins, and markers of oxidative stress. The negative for SARS-CoV-2 adolescents were recruited one month after the last molecular test. The control group were sera of age-matched healthy controls that never contracted the infection.
Interestingly, NGF serum was lower in post-infected-COVID-19 individuals when compared to healthy controls. BDNF and TGFb levels were higher concerning healthy individuals only in post-infected COVID-19 symptomatic and future long-COVID-girls, without other changes for this neurotrophin. Oxidative stress and inflammatory biomarkers were unchanged in both sexes.
They predicted that NGF and BDNF could be used as early biomarkers of COVID-19 morbidity in adolescents.
This manuscript is an original approach to markers for covid-19 and opens possibilities to measure these molecules at the beginning of the infection.
It would be interesting to know the BMI of the participants because adipose tissue also secretes NGF and BDNF, but maybe in a larger cohort.
I suggest the authors to speculate on the function of these markers in the infection, and the possible reason for the decrease in NGF in the girls, and the differences with the boys.
Reviewer 2 Report
the work by Petrella et al aims to assess biomarkers levels including serum NGF and BDNF in Long-COVID-19 Adolescents.
the work has some serious limitations pertaining to the cohort numbers they used as n=10 (5 male vs 5 females) which is not based on any sample calculation to reflect any power analysis; being said this work can be presented as a pilot study as it has extremely low N.
the cytokines levels should be correlated to clinical data that need to be tabulated and categorized in one table.
the work mentioned oxidative stress markers and then they mentioned that these are the different cytokines measured, thus, this needs to be corrected as oxidative stress markers are not assessed.
the discussion is lengthy and authors have over-described their results and discussed lung fibrosis etc.. and all this is based on blood cytokine levels. They should be more modest in their discussions s there were no clinical data or CT images of the lungs.
the paper is full of wrong English writing and requires an expert in the English editing
the abstract is lengthy and it is a copy of the introduction.
Reviewer 3 Report
The manuscript by Petrella et al. evaluates the association between serum Nerve Growth Factor (NGF), Brain-Derived Neurotrophic Factor (BDNF) and different sub-classes of COVID-19 symptoms in adolescents.
This is a case-control study with 30 COVID-19 cases divided into 3 symptoms subgroups (asymptomatic, acute symptomatic, future long-COVID-19) of 10 patients each (5 boys and 5 girls) and 10 healthy controls matched by sex and age, enrolled in a single hospital in Italy.
Study protocol design and description are poor. The study sample is very small and not sufficient to address the aim of the paper. Statistical analysis is partial and cannot lead to any evidence of notice.
The major limitations of the manuscript are:
- The sample size is small. Even if symptomatic COVID-19 in adolescents could be considered rare, at least the number of healthy controls should match the total number of COVID-19 cases.
- Statistical analysis is partial. The choice to perform the statistical analyses separating boys from girls leave each analyzed group with 5 individuals. At least the analyses of all COVID-19 cases vs controls should be presented. Moreover, there is a high probability that the associations presented in the results are casual, since having no clinical meaning at all. Correction for multiple testing should be applied.
- Results presentation. Since data presented in Figures 1-3 are numeric values divided by categories box-whiskers plots instead of bar plots should be used.